



# Spatially Compounded Surge Events: An Example from Hurricanes Matthew and Florence

Scott Curtis[1], Kelley DePolt[2], Jamie Kruse[3], Anuradha Mukherji[2], Jennifer Helgeson[4], Ausmita Ghosh[3], and Philip Van Wagoner[2]

[1]Department of Physics and Lt. Col. James B. Near, Jr., USAF, '77 Center for Climate Studies, The Citadel, Charleston, SC, 29409, USA
[2]Department of Geography, Planning and Environment, East Carolina University, Greenville, NC, 27858, USA
[3]Department of Economics, East Carolina University, Greenville, NC, 27858, USA
[4]Engineering Laboratory, National Institute of Standards and Technology, Gaithersburg, MD, 20899, USA

*Correspondence to*: Scott Curtis (wcurtis1@citadel.edu)

**Abstract.** The simultaneous rise of tropical cyclone induced flood waters across a large hazard management domain can stretch rescue and recovery efforts. Here we present a means to quantify the connectedness of maximum surge during a storm with geospatial statistics. Tide gauges throughout the extensive estuaries and barrier islands of North Carolina deployed and operating during Hurricanes Matthew (n=82) and Florence (n=123) are used to compare the spatial compounding of surge for these two disasters. Moran's I showed the occurrence of maximum storm tide was more clustered for Matthew compared to Florence, and a semivariogram analysis produced a spatial range of similarly timed storm tide that was four times as large for Matthew than Florence. A more limited data set of fluvial flooding and precipitation in eastern North Carolina showed a consistent result – multivariate flood sources associated with Matthew were more concentrated in time as compared to Florence. Although Matthew and Florence were equally intense, they had very different tracks and speeds, which influenced the timing of surge along the coast. We hope this method could be used for other landfalling tropical cyclones to better understand the drivers that lead to spatially compounded surge events.

# 1 Introduction

## 1.1 Spatially Compounded Weather Events

Compound climate and weather events have been defined as the "combination of multiple drivers and hazards that contributes to societal or environmental risk" (Zscheischler et al., 2018), undermining hazard management (Raymond et al., 2020). For example, emergency managers and planners often use tools that only consider one hazard to estimate risk, whereas the combination of hazards leads to a nonlinear increase in risk (Moftakhari et al., 2017). This underestimation of risk leads to under-resourcing for prevention prior to an event and the adoption of extra measures for recovery once the risk is realized. Also, the complex nature of compound hazards in time and space during an event may stretch first responders to capacity. Thus, the entire emergency management cycle of response, recovery, mitigation and preparedness needs to be reframed in the context of compounded hazards (Raymond et al., 2020).



Zscheischler et al. (2020) expanded the typology of compound weather and climate events to include four themes: preconditioned, multivariate, temporally compounding, and spatially compounding. They also include a modulator to the
definition, which is a state of the atmosphere or climate, such as a climatic teleconnection, stationary weather pattern, or storm that increases the probability of the drivers. This study adds to our understanding of the spatially compounding event type, which Zscheischler et al. (2020) define as "multiple-connected locations affected by the same or different hazards, within a limited time window, thereby causing an impact". Spatially compounded events can restrict emergency response actions if the operational space (jurisdiction) experiences coincidental hazards (Leonard et al., 2014). This can lead to a disaster
situation, as about 50% of deaths typically occur within a few hours after an event (Narayanan and Ibe, 2012). Thus, the temporal component is an important consideration.

Several studies have used spatial statistics to better understand compounded hazards, mostly on a national scale. For example, Touma et al. (2018) used the semivariogram to characterize length scales of extreme daily precipitation ($90^{th}$ percentile) in the
US. This analysis was refined to determine the spatial extent of extreme precipitation during the evolution of landfalling tropical cyclones (Touma et al., 2019). Du et al. (2020) used a rotating calipers algorithm to quantify the spatial scales of heavy Meiyu precipitation events, and Blanchet and Creutin (2017) modelled the probability of co-occurrence of extreme precipitation in the French Mediterranean region with the Brown-Resnick max-stable process. More recently the connectedness of fluvial flooding has been examined using the F-madogram (Brunner et al., 2020). While these studies provide
important climatological information that will inform pluvial and fluvial flood risk, the current study analyzes another flood hazard – storm surge – and defines the spatially aggregate impacts within an individual storm.

Storm surge – or the rise of coastal waters generated by storm winds – is the greatest threat from tropical cyclones globally (NHC 2020). In the US this hazard amounted to almost half of hurricane fatalities between 1963 and 2012 as people either
could not or would not evacuate (Rappaport, 2014). Evacuation can become problematic in a spatially compounded event with the simultaneous inundation of roadways and railways within a region (Koks et al., 2019; Zscheischler et al., 2020). Two contrasting cases will be explored, both of which had a severe impact on the US: Hurricane Matthew, which made landfall south of McClellanville, SC as a category 1 storm at 1500 UTC on October 8, 2016; and Hurricane Florence, which made landfall approximately 250 km to the northeast at Wrightsville Beach, NC, also a category 1 storm at the time, at 1115 UTC
on September 14, 2018. Eastern North Carolina (Figure 1) is the focus of our study for a number of reasons. Firstly, Matthew and Florence caused more direct fatalities in North Carolina than any other state. Secondly, North Carolina has a large estuary system and extensive inhabited barrier island complex comprising 17,152 km of shoreline, which complicates evacuation and rescue operations during storm surge, but is conducive to a spatial analysis. Finally, this study is part of a larger project related to compound flood risk and management in rural eastern North Carolina.




## 1.2  Hurricanes Matthew and Florence

According to the National Hurricane Center's cyclone reports, Hurricanes Matthew and Florence were the 10[th] and 9[th] most destructive storms in the US (Stewart, 2017; Stewart and Berg, 2019).  In North Carolina alone, Matthew caused $1.5 billion in property damage, while Florence caused $22 billion.  However, Matthew caused more direct fatalities in the state (25) as compared to Florence (15).  Casualties and damages were primarily attributed to pluvial and fluvial flooding and storm surge. Thus, these storms would be appropriately categorized as multivariate compound events (Zscheischler et al. 2020).  However, Zscheischler et al. (2020) recognize that their defined typology has soft boundaries and that events can fall into multiple categories.  Figure 2a shows the track of Hurricane Matthew.  The storm paralleled the study area from 1800 UTC October 8, 2016 and exited off the coast of North Carolina around 0900 UTC October 9 (~ 15 hours in total).  During this time much of the coast experienced winds greater than 50 kts.  Matthew's highest recorded storm surge in North Carolina was 1.76 m above mean highest high water (MHHW) on Hatteras Island (Stewart, 2017).  Hurricane Florence's track (Fig. 2b) was nearly perpendicular to Matthew's as the storm approached the coast at 0000 UTC September 14, 2018 from the east and exited the study area around 0000 UTC September 15 (~ 24 hours in total).  Wind speeds diminished from hurricane force in the south to tropical storm force in the north.  Florence's highest recorded storm surge in North Carolina was 3.17 m above MHHW near New Bern (Stewart and Berg, 2019).

## 2.  Data and Methods

Storm tide data for Hurricanes Matthew and Florence were collected from the US Geologic Survey Flood Event Viewer (https://stn.wim.usgs.gov/FEV/).  Flood Event Viewer provides a file of locational data for all tidal stations – rapid deployment and pressure transducers - in operation during significant weather events.  For Hurricane Florence a peak summary file was also available, which includes the unfiltered peak water level recorded in feet above NAVD88 and the time of the recording. Matthew did not have such a summary, so these attributes were manually recorded from each sensor's page and appended to the locational data.  Heights were converted to meters and data from outside North Carolina was removed.  Some of the stations in the peak summary file had an initial storm tide related value and a subsequent river runoff value.  Only the surge information was retained.  The final number of peak storm tides in North Carolina after a quality control was 82 for Matthew and 123 for Florence.

To understand how an event is spatially compounded requires either quantifying the spatial extent over a predefined time interval or the temporal extent over a predefined spatial domain.  An example of the former definition would be delineating roadways impacted by flooding over the duration of the storm.  However, characterizing the capacity of emergency responders to perform their job within a set jurisdiction would fall under the later definition.  It is argued that if peak storm tide occurs simultaneously within the jurisdiction it would stretch the ability of search and rescue teams, such as Swiftwater, part of the North Carolina Division of Emergency Management (NCDEM).  The tide gauge data collected for both storms falls under the



Eastern Branch of the NCDEM (see Fig. 1 for boundary). It is assumed that the geographic extent of surge flooding will be greatest at the time the tide gauge is at its maximum. To compare the temporal evolution of peak surge across storms requires a reference time. Here we selected the time of landfall: 0800 UTC October 9, 2016 for Matthew and 1115 UTC September 14, 2018 for Florence. Differences between time of peak surge and landfall ($\Delta t_{ps\text{-}lf}$) were calculated in days and added to the peak surge tables.


A Kolmogorov-Smirnov test was used to determine if the distributions of tide magnitude and timing ($\Delta t_{ps\text{-}lf}$) were significantly different between Matthew and Florence. The D statistic is calculated as the maximum difference between two cumulative distributions, $S_{N_1}(x_i)$ and $S_{N_2}(x_i)$:

$$D = \max_{1 \leq i \leq n} |S_{N_1}(x_i) - S_{N_2}(x_i)|. \tag{1}$$


Next, two tests were performed in ArcGIS to understand the spatial correlation of $\Delta t_{ps\text{-}lf}$: Global Moran's I and the range of the semivariogram function. Moran's I is an inferential statistic, where the null hypothesis is that a variable x, here x = $\Delta t_{ps\text{-}lf}$, is randomly distributed throughout a spatial range:

$$I = \frac{n}{S_o} \frac{\sum_{i=1}^{n} \sum_{j=1}^{n} w_{i,j} z_i z_j}{\sum_{i=1}^{n} z_i^2}, \tag{2}$$

where z is the deviation of x from its mean, $w_{i,j}$ is the spatial weight between $x_i$ and $x_j$, and n is the total number. $S_o$ is the sum of all spatial weights:

$$So = \sum_{i=1}^{n} \sum_{j=1}^{n} w_{i,j}. \tag{3}$$

An expected Moran's I is computed and compared to the observed value to generate a z-score and p-value for significance testing. A significantly positive (negative) value of Moran's I would suggest that the variable x is clustered (dispersed). A

Moran's I test has been used previously to determine the spatial clustering of precipitation within the Gulf of Mexico during phases of the El Nino/Southern Oscillation (Munroe et al. 2014). Next, we follow the procedure of Touma et al. (2018, 2019) and use the semivariogram to test the hypothesis that variables closer in space would be more similar than those far apart. Thus, the raw semivariogram is a function of distance h between variables $x_i$ and $x_j$, and is simply half the variance or:

$$\gamma(h) = \frac{\overline{(x_i - x_j)^2}}{2} \tag{4}$$

The data can then be modelled statistically with a family of functions. Here we choose the stable variogram function:

$$\gamma_{sta}(h) = b + C_o * (1 - e^{-\frac{h^s}{a}}) \tag{5}$$

where $C_o$ is the sill of the variogram, s is the shape parameter, a is the range, and b is the nugget. These parameters are graphically represented in Figure 3. In this study we are particularly interested in the range, a, as this is when the spatial autocorrelation first becomes negligible. Thus, the range gives an indication of spatial scale for coincidental storm surge.




## 3. Results

Table 1 gives some basic statistics for storm surge peak magnitude. In this data set Matthew had the largest storm tide at 5.08m recorded at a pressure transducer (NCDAR12768) at the Sanderling Resort (Atlantic coast of Dare County). However, Florence had more outlier peak storm tides in excess of 3.5m. Both storms had similar means and standard deviations. The

D statistic between the two storms was 0.175, which did not reach the 0.05 significance level, so the distributions cannot be considered different (Table 1).

Interestingly, the distribution of $\Delta t_{ps\text{-}lf}$ was very different between Matthew and Florence (Table 2). Both storms had mean peak storm tide at less than one day after landfall (10-14 hours). However, the standard deviation of $\Delta t_{ps\text{-}lf}$ was less than the

mean for Matthew (0.34 days), but more than double the mean for Florence (1.29 days). There were seven outliers in the Florence record from 3.53 days to a maximum of 7.33 days. It is not surprising then that the D statistic between the two storms was 0.309 and highly significant. Histograms of $\Delta t_{ps\text{-}lf}$ in 15-minute increments are given in Figure 4a. As suggested from Table 2, Matthew's peak storm tide was clustered in time, with 66% occurring within the first 12 hours after landfall. Only 37% of Florence's peak surge occurred during this critical time.


Complementary to the difference in the distribution of $\Delta t_{ps\text{-}lf}$ between Matthew and Florence, the Moran's I test (Table 3) shows that for Matthew $\Delta t_{ps\text{-}lf}$ was significantly clustered in space, equating to a spatially compounded event, whereas the insignificance of Moran's I for Florence indicates random spatial distribution. Values of $\Delta t_{ps\text{-}lf}$ along coastal North Carolina for the two storms are shown as circles in Figure 5. For Matthew there is very little difference in $\Delta t_{ps\text{-}lf}$ geographically although

some of the later values appear to occur on the Outer Banks in Dare county (see Fig. 1 for reference). In the case of Florence, from pre-landfall to 1.5 days after there appears to be a north to south gradient in $\Delta t_{ps\text{-}lf}$. The earliest peaks, taking place before landfall, occurred on the Outer Banks and the mid-region of Beaufort, Pamlico and Carteret counties (see Fig. 1 for reference). Peaks that occurred between 0 and 1.5 days after landfall are generally found along the southeastern coast – Onslow, Pender, New Hanover, and Brunswick counties. However, peaks occurring 1.5 to 2.5 days after landfall appear in the north region of

Bertie, Washington and Tyrrell counties with some values near Cape Hatters in Dare county. $\Delta t_{ps\text{-}lf}$ on the order of 3-4 days can be found even further north in Pasquotank, Pender, and Currituck counties. Finally, the maximum $\Delta t_{ps\text{-}lf}$ of over 7 days is seen in New Hanover county.

Matthew and Florence can both be considered multivariate compound flood events, which not only produced surge flooding,

but also pluvial and fluvial flooding. However, real-time data collection from stream and precipitation gauges during the storms were more limited in the coastal environment: four stream gauges (boxes in Figure 5) and two rain gauges (Xs in Figure 5) were analysed to complement the more comprehensive surge analysis. Figure 4b (Figure 4c) shows 15-minute precipitation data during Hurricane Matthew (Florence) and the time when the stream gauges first reached minor flood stage. The time





scales were adjusted to match Figure 4a. Lake Mattamuskeet near Fairfield, NC (northern X in Figure 5) had higher rainfall totals for Matthew (166.8 mm) compared to Florence (103.1 mm), with a maximum rain rate of 15.2 mm in 15 minutes (~61 mm hr$^{-1}$). Heavy rainfall, defined as >2.5 mm in 15 minutes or >10 mm hr-1 (Met Office, 2011) was first observed at the beginning of Matthew's period of record (POR) and last observed 15.25 hours after landfall, both at Lake Mattamuskeet. Interestingly, for both storms Cape Fear Lock #1 near Kelly, NC (southern X in Figure 5) had the highest rainfall totals, recording 204 mm for Matthew and an astounding 622 mm of rainfall during Florence. This hurricane was a massive rainstorm in the southeastern portion of the study area. However, Florence's first instance of heavy rainfall occurred at Lake Mattamuskeet at the beginning of the POR. Lock #1 recorded its first instance of heavy rainfall 4.75 hours prior to landfall and its last over 3.5 days after landfall, reaching a peak of 21.3 mm of rain in 15 minutes (85.3 mm hr$^{-1}$). Most of the stream gauges for both storms reached minor flood stage from landfall to 1.4 days after (Fig. 4b,c). However, for Florence, three streams reached minor flood stage prior to landfall: Swift Creek near Streets Ferry, NC, Trent River near Pollocksville, NC, and Pamlico River at Washington, NC (see squares in Fig. 5b). Also, during Florence Lock #1 (same location as rain gauge) reached flood stage 3.2 days after landfall (Fig. 4c). Thus, the precipitation, stream and tide data paint a consistent picture that Matthew's flooding across eastern North Carolina was more synchronised as compared to Florence.

In the final part of the analysis, semivariograms are used to determine a spatial scale of the storm tide hazards for Matthew and Florence. Table 3 gives the parameters of the stable semivariogram functions and Figure 6 shows the raw and modelled data. Since the variance of the difference in x (γ) increases with distance, the semivariogram can be considered a dissimilarity function. Florence has much larger values of γ than Matthew, even with small distances. For the stable model, Florence reaches a sill of 1.33 at a range of 25.5 km, while Matthew reaches a sill of 0.05 at a range of 53.5 km. The shape parameter of Matthew is less than 2, which means the stable semivariogram resembles an exponential curve more than the case of Florence. The range gives a quantification of the length scale of timing of peak storm tide. In other words, for the case of Matthew (Florence) one would need to be 53.5 km (25.5 km) distant from a given tide station to reach a dissimilar tide station in terms of $\Delta t_{ps\text{-}lf}$ . This range is graphically represented by the yellow circles on Figure 5. For Matthew the hazard zone has an area of 8,997 km$^2$, and is roughly the size of three eastern North Carolina counties, whereas the Florence hazard zone is less than one quarter of the size at 2,051 km$^2$.

## 4. Discussion and Conclusion

This study contributes to our understanding of a spatially compounded event or "multiple-connected locations affected by the same or different hazards, within a limited time window, thereby causing an impact" (Zscheischler et al., 2020). In order to account for the "multiple-connected locations" and "limited time window", the time of hazard is analysed geospatially, within a predetermined region. We apply this definition to storm tide, specifically the peak height recorded, during Hurricanes Matthew and Florence for coastal North Carolina. A Moran's I test was used to determine whether there was a clustering of coincident surge hazards and a semivariogram analysis was used to model the relationship between time of peak surge and





distance. The first peak surge during Matthew occurred 15 minutes prior to landfall and the last one occurred 28.5 hours after landfall. In contrast the first peak surge during Florence occurred 22.25 hours before landfall and the last one occurred over seven days after landfall. It is not surprising then, that both statistical tests definitively show that Matthew's peak surges

happened more simultaneously than Florence. Furthermore, river flooding and extreme rainfall also occurred within a narrower time interval for Matthew, as compared to Florence. One important output from the semivariogram analysis is the range or length scale of similarly timed peak surge. For Matthew this "area of effective hazard zone" was four times as large as Florence. This suggests that given the same resources, there would be a greater risk of isolation, injury, and death in coastal communities due to surge from Matthew as compared to Florence. However, it should be noted that there were no direct

deaths associated with storm surge from either storm. According to the Associated Press (2018) North Carolina Governor Cooper noted that twice as many people were saved from rising floodwaters during Florence as compared to Matthew. This could partially be due to the timing of the water hazards, but more likely it is due to the ever-increasing preparedness for storms by the NCEM, Swiftwater, and other related agencies.

In summary, this work adds to previous compound hazard studies by applying conventional geospatial analysis techniques to surge within a storm environment. However, there are important caveats to this study that need mention. First, because Matthew was the more spatially compounded storm, does not necessarily mean it was the more devastating surge event. In fact, one could argue that because Florence was long lasting and affected different regions of the coast at different times, it was more difficult to manage. At the same time, a more spatially compounded event would make a coordinated response

effort and/or reciprocity across counties more challenging. Second, we didn't consider the spatial extent of surge flooding, only the tide gauge locations. Schaffer-Smith et al. (2020) found the total flood extent was similar between the two storms in North Carolina, with Florence causing more extensive flooding in the southeastern part of the state. Third, this is a purely statistical study and does not offer reasons for the difference in spatial compounding between Matthew and Florence. However, it is clear that many characteristics of the two storms were quite different as presented in Figure 2. Unlike Matthew, Florence

moved slowly westward so that portions of the coast received onshore winds for an extended period of time covering several tide cycles. In the definition of a compound hazard, size, track, speed, tides, etc. would be considered the drivers and the storm would be the modulator (Figure 7). Finally, this study motivates future work to consider a climatology of storm characteristics for developing relationships between the compound hazard drivers and the spatial metrics of surge presented here. Operationally, forecasted storm parameters could then be used in emergency response preparation, with the level of spatial

compounding informing the number and distribution of response teams.



**Data availability.**

The data used in this paper can be requested from the corresponding author. We ask interested researchers to please contact the corresponding author of this article.


**Author contributions.**

SC collected and analysed the data, created the tables and figures, and wrote the initial draft. KD, JM, AM, JH, AG, and PV contributed to the interpretation of the results and reviewed the manuscript.

**Competing Interests.**

The authors declare they have no conflict of interest.

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



Table 1. Basic statistics on North Carolina peak storm tide during Hurricanes Matthew and Florence. A measure of the difference in the distributions is given with the Kolmogorov-Smirnov D-statistic. Significance level is denoted by the p-value.


| Storm | n | Mean storm tide (m) | Standard deviation | Max, min | Outliers | D-statistic K-S test | p-value |
|--------|-----|-----|-----|------|------------------|-------|-------|
| Matthew | 82 | 1.79 | 0.79 | 5.08, 0.66 | 5.08 | | |
| Florence | 123 | 1.86 | 0.80 | 3.75, 0.54 | 3.52, 3.59, 3.65, 3.75 | 0.175 | 0.088 |



Table 2. Basic statistics on the difference in time (days) between peak storm tide in North Carolina and landfall ($\Delta t_{ps-lf}$) for Hurricanes Matthew and Florence. A measure of the difference in the distributions is given with the Kolmogorov-Smirnov D-statistic. Significance level is denoted by the p-value.


| Storm | n | Mean $\Delta t_{ps-lf}$ (day) | Standard deviation | Max, min | Outliers | D-statistic K-S test | p-value |
|--------|-----|-----|-----|------|------------------------------|-------|-------|
| Matthew | 82 | 0.40 | 0.34 | 1.18, -0.02 | NA | | |
| Florence | 123 | 0.59 | 1.29 | 7.33, -0.93 | 3.53, 4.10, 4.18, 4.24, 4.27, 4.59, 7.33 | 0.309 | 0.000 |








Table 3.  Moran's I statistic and associated p-value for $\Delta t_{ps\text{-}lf}$ for Hurricanes Matthew and Florence.

| Storm | n | Moran's I | p-value |
|---|---|---|---|
| Matthew | 82 | 0.835 | 0.000 |
| Florence | 123 | 0.085 | 0.523 |


Table 4.  Statistics of the stable semivariogram models (see Figure 6) of $\Delta t_{ps\text{-}lf}$ for Hurricanes Matthew and Florence.

| Storm | Sill (Co) | Shape (s) | *Range (a)* | Nugget (b) |
|---|---|---|---|---|
| Matthew | 0.05 | 0.791 | *53,514 m* | 0.011 |
| Florence | 1.33 | 2 | *25,548 m* | 0.432 |







Figure 1. Orientation map of North Carolina counties that fall within the Eastern Branch of the North Carolina Department of Emergency Management. © OpenStreetMap contributors 2020. Distributed under a Creative Commons BY-SA License.


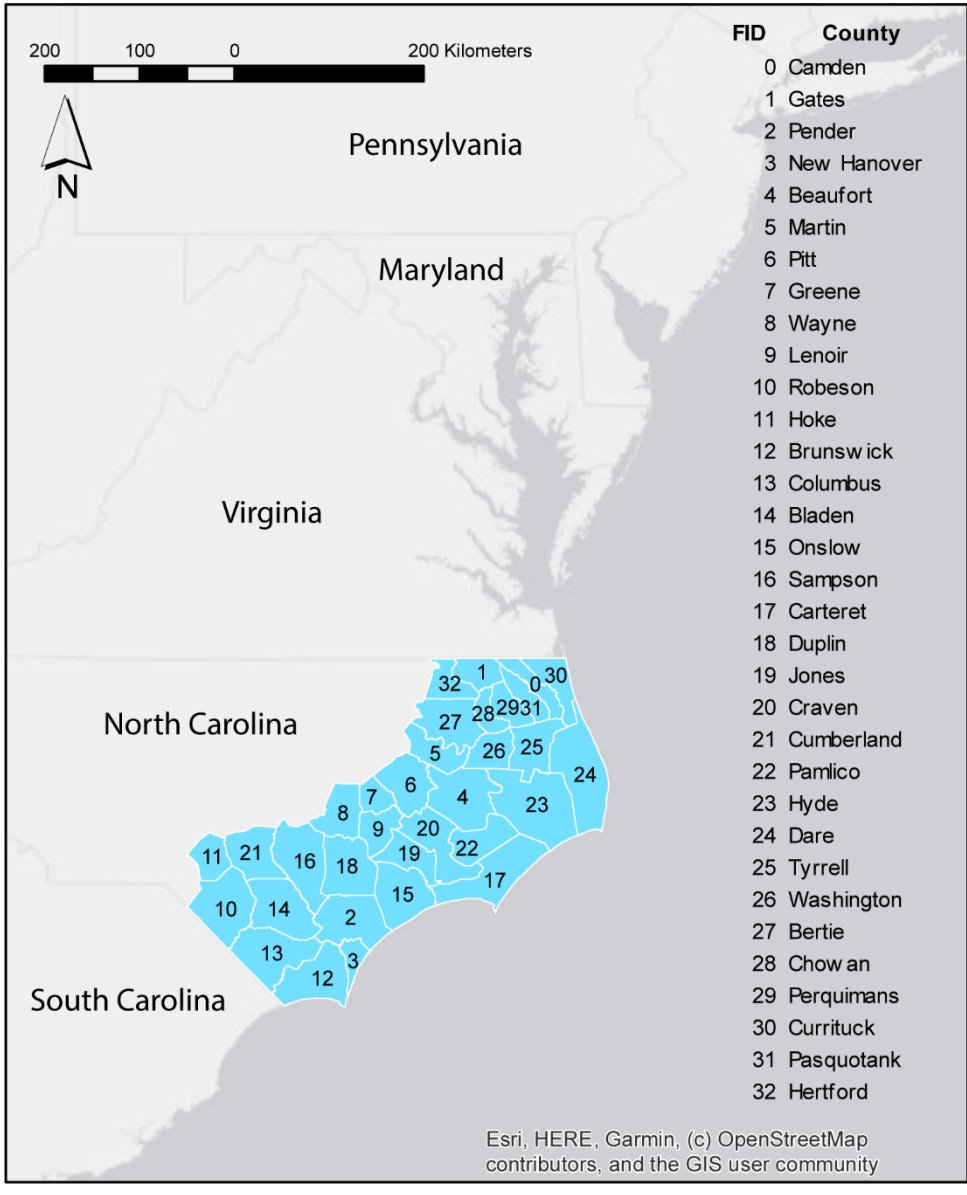




Figure 2. Track and wind speeds during the passage of Hurricanes (A) Matthew and (B) Florence off the coast of North Carolina. Red stippling indicates hurricane force winds (> 33 ms$^{-1}$), orange storm force (> 26 ms$^{-1}$) and green tropical storm force (> 17 ms$^{-1}$). Track is denoted by dashed line. First number is the day and the second number is the time (UTC) in October 2016 for (A) and September 2018 for (B)


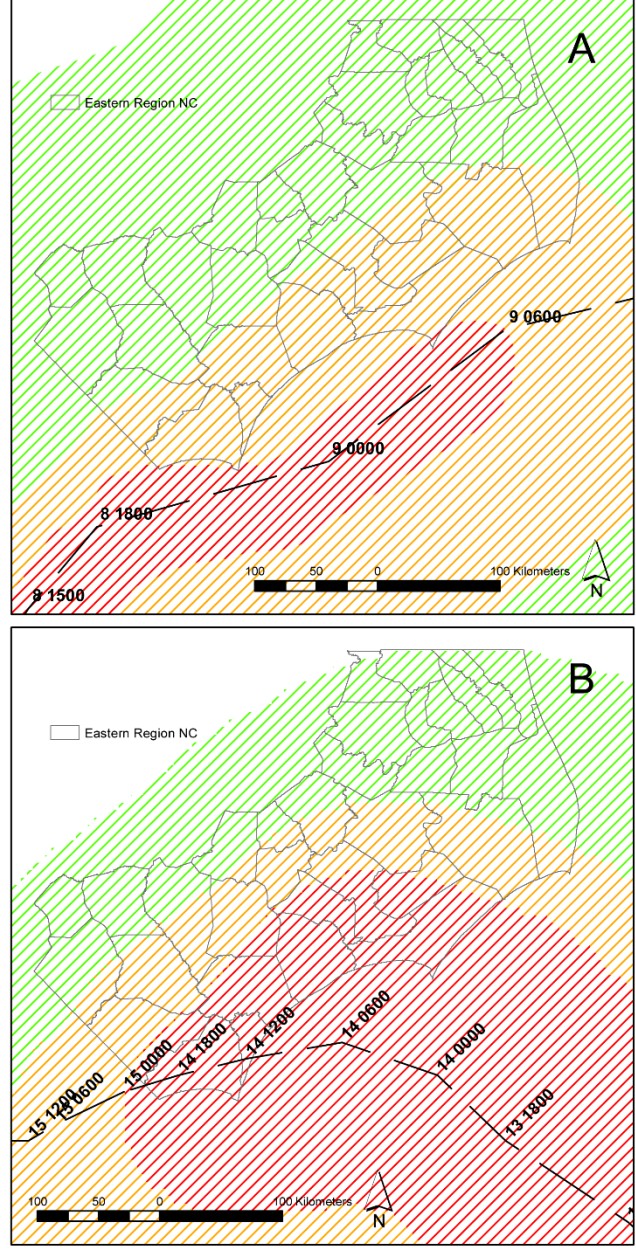




Figure 3. Schematic of a semivariogram γ as a function of distance (h) with the nugget, sill, and range defined.

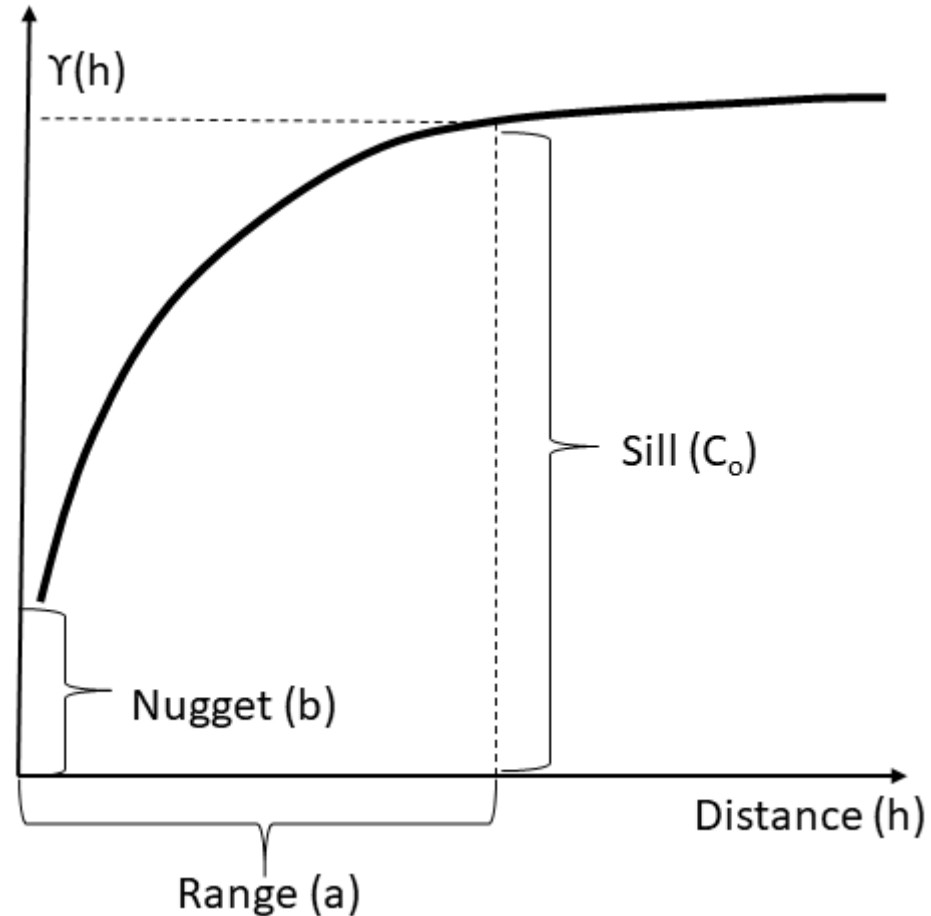








Figure 4 (next page). Time evolution of surge, rainfall, and river flooding during Hurricanes Matthew and Florence in North Carolina. X-axis is $\Delta t_{ps-lf}$ binned in 15-minute increments (unit days) and covers approximately one day prior to landfall to seven days after A) percent of peak surge occurrences; blue is for Matthew and red is for Florence. B) precipitation (mm)
during Matthew; blue bars represent rainfall recorded at the Cape Fear Lock #1 near Kelly, NC and black bars represent rainfall recorded at Lake Mattamuskeet near Fairfield, NC. Crosses indicate when nearby river gauges first reached minor flood stage. Bold cross represents two river gauges reaching minor flood stage in the same 15-minute increment. C) precipitation (mm) during Florence; red bars represent rainfall recorded at the Cape Fear Lock #1 near Kelly, NC and black bars represent rainfall recorded at Lake Mattamuskeet near Fairfield, NC. Crosses indicate when nearby river gauges first reached minor flood stage.
Bold cross represents two river gauges reaching minor flood stage in the same 15-minute increment.

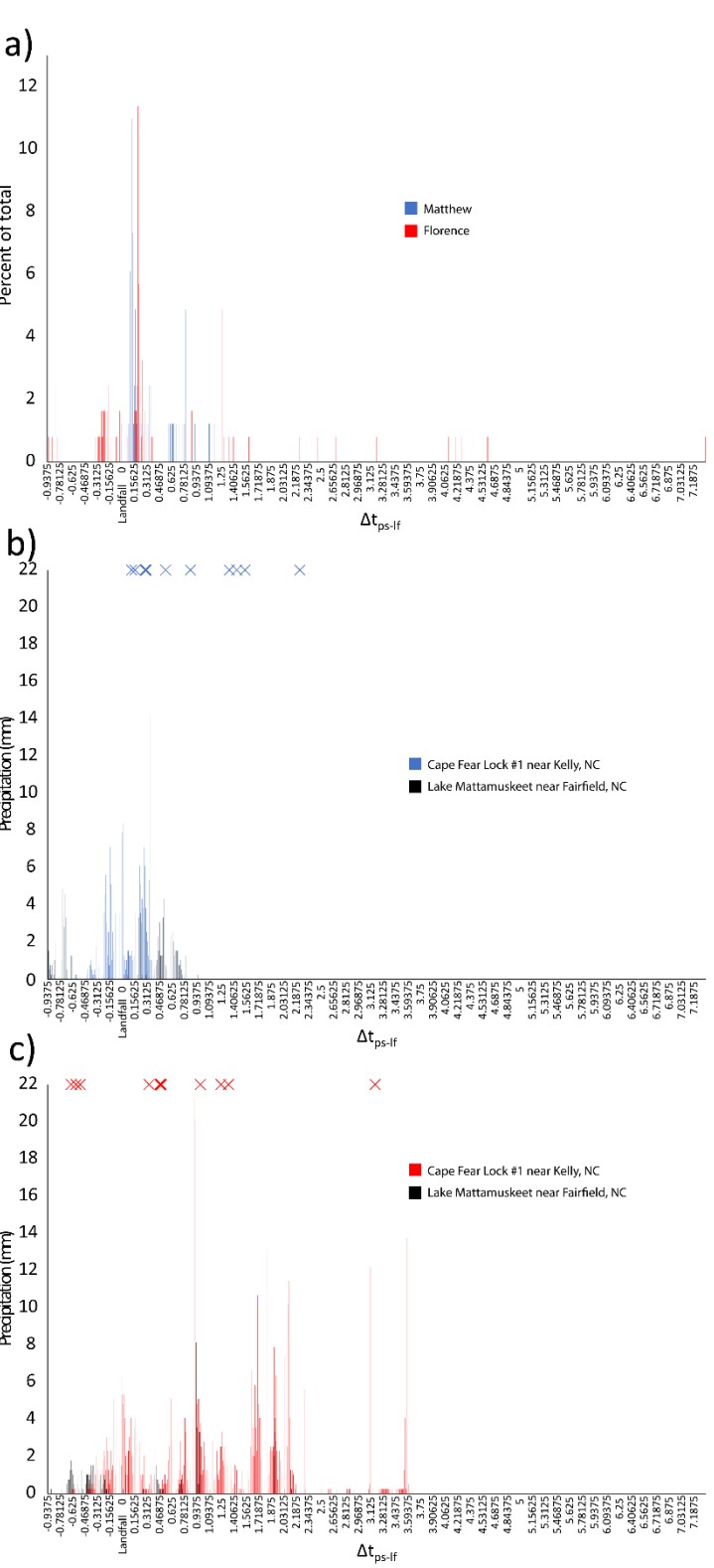


Figure 5. Maps of $\Delta t_{ps-lf}$ (circles), time when river gauges first reached minor flood stage (squares) and locations of rain gauges (crosses). Large circles represent the area of similarity in $\Delta t_{ps-lf}$ computed using the semivariogram range (a) as the radius. A) is for Matthew and B) is for Florence.

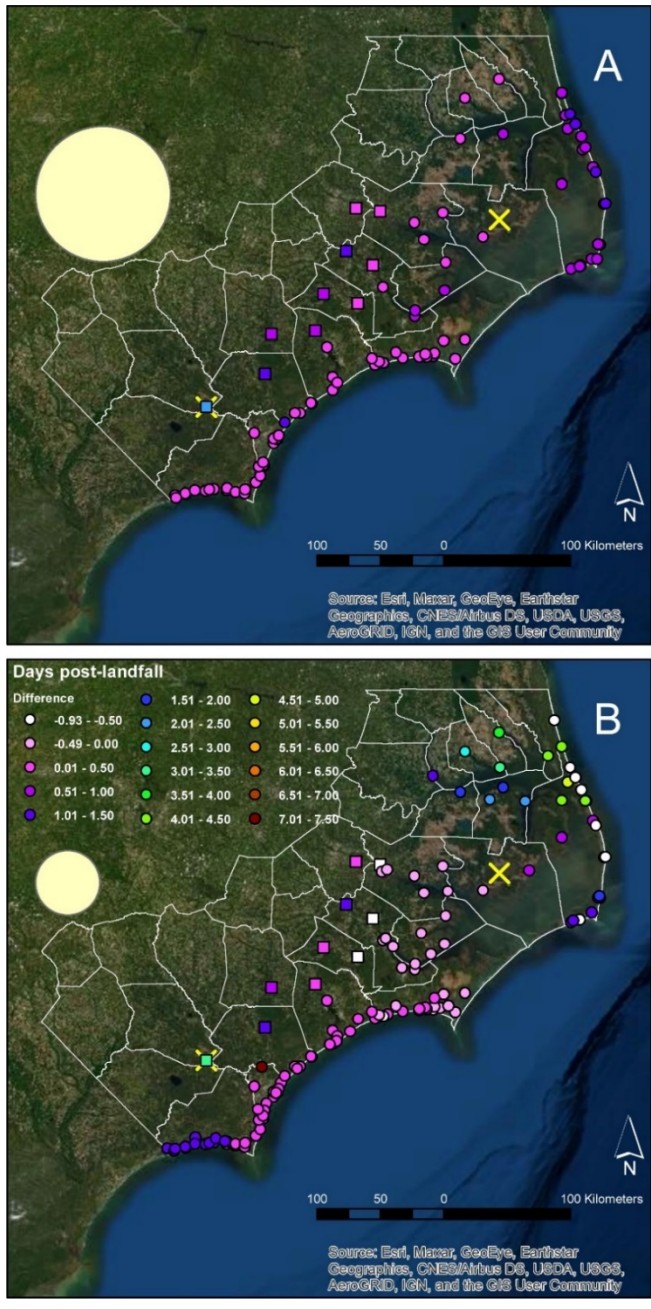



Figure 6. Observed values (points) and stable semivariogram models (curves) for γ(h). X-axis is in meters. Blue represents Matthew and red Florence. Note that only γ(h) < 2.0 are displayed.






Figure 7. Elements that make up a spatially compounded surge hazard, as described in this study (see Zscheischler et al. 2020).

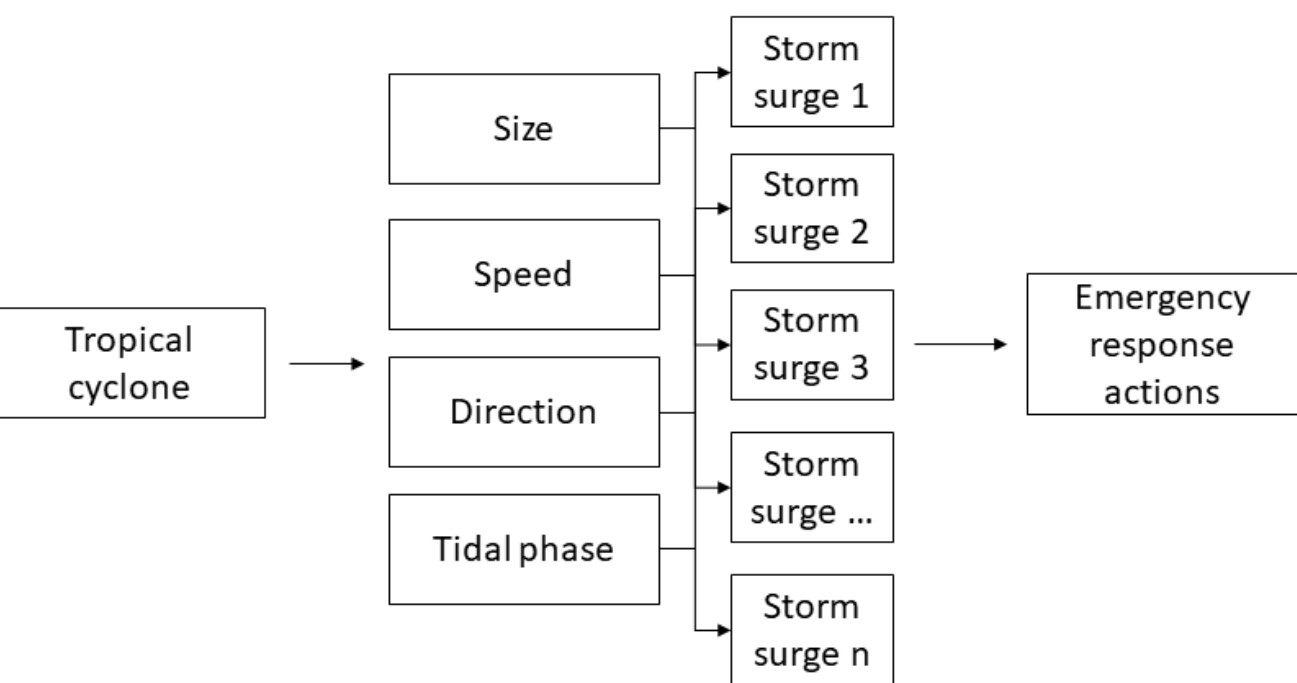
