# Peer review of "Spatially Compounded Surge Events: An Example from Hurricanes Matthew and Florence"

_Natural Hazards and Earth System Sciences, 2021_

## Author Response (AR1)

** Please note that in the Track Changes version of our paper the Tables are not appearing correctly. However, we did not make any modifications to the Tables and they appear correctly in the final version.

**Reviewer 1**

*Just a couple minor comments: 1. The goal of the study is somehow not well stated in the introduction. 2. On the abstract, the last sentence belongs more in the conclusions, and not in the abstract. 3. L52 add "and severe drop in atmospheric pressure" after winds.*

1. Thank you for your comments. We agree that our goal of characterizing the spatially compounded nature of storm surge was not well articulated in the introduction and this has been remedied.
2. The reviewer is correct that the last sentence does not belong in the abstract and has been removed.
3. Secondary to winds, a severe drop in atmospheric pressure contributes to storm surge and this has been added as suggested.

**Reviewer 2**

*General comments*

*This paper considers the spatial compounding of storm surge in Eastern North Carolina during Hurricanes Matthew and Florence. While the study provides a useful method for understanding the spatial extent of surge hazards and informing emergency management activities, I am not convinced that all of the data presented actually represent wind-driven storm surge. Hurricane Florence's winds dropped to 45mph by the end of the day on September 14, and the storm appears to have left the study area by September 15. This suggests to me that the peak water levels associated with Δt>3 in the northern part of the study area in Figure 5 are actually driven by another process, not by wind-driven surge. This would likely mean that the time distributions of the two hurricanes are actually more similar than reported here.*

*Specific comments*

*Line 106: It is unclear to me what distribution the K-S test was applied to. The text says it was used to evaluate the distributions of tide magnitude and timing – was this done at each location within the study site, or was the spatial distribution compared at each time after landfall? This is clarified a bit in Tables 1-2 but should be explained in the Methods.*

*Line 125: The authors never explain why the stable variogram was chosen. Were other variograms tested? The results of the model testing should be presented, if not in the paper then in a supplement.*

*Line 115: "n is the total number" of what?*

*Figure 3: I don't think is it necessary to show an example of a semivariogram. The authors could point out these features on the fitted semivariograms in Figure 6.*

*Figure 4: Do the crosses indicate that rivers near Cape Fear reached flood stage during both storms, but no rivers near Lake Mattamuskeet ever reached flood stage (since there are no black crosses)? Please explain what "near" means in the caption (near to what?).*

*Figure 6: These data should be plotted on separate graphs so that the semivariogram for Matthew can be seen more clearly.*

*Technical corrections*

*Figure 2: The day/time text is difficult to read because it overlaps the storm track line and other text. Please adjust so this information is more legible.*

*Wind speeds are reported in knots in the text but m/s in Figure 2. Please be consistent. Line 161: Replace "boxes" with "squares" to be consistent with the Figure 5 caption. Line 181: This reference should be for Table 4.*

General comments.

We appreciate the reviewer's observation and we were remiss in describing the evolution of Hurricane Florence from September 15 to 18.  During that time Florence recurved to the north and by 5am on the 18th the storm was classified as extratropical and positioned at 41.3N and 75.9W (see daily weather map below).  At the time many of the northern coasts of the northern estuaries (Albemarle Sound) were receiving significant SE winds, lowest pressure, and highest surge.  An example is given for NCCUR12568. This important information is now included in the manuscript.

[Figure]

Surface Weather Map and Station Weather at 7:00 A.M. E.S.T.

[Figure]

[Figure]

[Figure]

Specific comments.

Line 106: The methodology has been clarified.

Line 125: There are several semivariogram models to choose, but over half are inappropriate for the data. For example, the exponential semivariogram, used by Touma et al. (2018), gives an unphysical range for Florence (see Table A below). After a visual inspection and some experimentation, we selected the stable semivariogram. By in large, the results from other candidate models yield the same key result – Matthew has a larger range than Florence. We can add more information as a supplement if necessary.

Table A. Statistics of the exponential semivariogram models of $\Delta t_{ps\text{-}lf}$ for Hurricanes Matthew and Florence.

| Storm | Sill (Co) | Shape (s) | *Range (a)* | Nugget (b) |
|---|---|---|---|---|
| **Matthew** | 0.047 | N/A | *36,877 m* | 0.012 |
| **Florence** | 0.894 | N/A | *1,369 m* | 0.0 |

Line 115: We apologize for the omission. It should be number of tide gauge stations.

Figure 3: We have removed Figure 3 and point out these features in Figure 6.

Figure 4: The crosses are representative rivers in the Eastern Branch of the NCDEM that were affected by the two hurricanes (squares in Figure 5). Only the Cape Fear Lock #1 location contains both a river gauge and a rain gauge. The coloring of the crosses and labeling probably adds to the confusion. The figure and table caption have both been revised and the accompanying text clarified.

Figure 6: We have plotted the graphs separately.  However, one reason to plot them on the same scale is to emphasize that Florence has much larger values of γ than Matthew indicating that Florence was more "dissimilar" than Matthew even at short distances between tide gauges.

Technical corrections

We have addressed all the technical corrections.

---

## Author Response (AR2)

We appreciate Reviewer 2's comments and have changed the text accordingly.

1. With regards to describing Florence's path earlier in the manuscript, we've replaced the following text on page 3:

*Florence turned northward and became a post—tropical cyclone.  The last National Hurricane Center report located its center at 42.6N and 71.9W at 1500 UTC September 18.  During Florence's passage through the study area wind speeds diminished from hurricane force in the south to tropical storm force in the north.*

with

*During Florence's passage through the study area wind speeds diminished from hurricane force in the south to tropical storm force in the north (Fig. 2b).  Florence then turned northward and became a post—tropical cyclone.  The last NHC report located its center at 42.6N and 71.9W (near Boston, MA) at 1500 UTC September 18.  The storm was elongated so at this time south-easterly winds were observed in the northern portion of the study area.*

2. With regards to the choice of variogram we've replaced the following text on page 4-5:

*The data can then be modelled statistically with a family of functions.    After some experimentation and visual inspection we decided to use the stable variogram function … where $C_o$ is the sill of the variogram, s is the shape parameter, a is the range, and b is the nugget.*

with

*The data can then be modelled statistically with a family of functions.  The most appropriate fitted semivariogram for our data was provided by the stable model: … where $C_o$ is the sill of the semivariogram, s is the shape parameter, a is the range, and b is the nugget.  We experimented with other functions, but it was discovered that over half produced unphysical results, while the others yielded the same basic results as the stable model.  An added advantage of the stable model is the shape parameter, s, which transforms the function to be more like an exponential or spherical model when s <= 2 and more like a Gaussian model when s > 2.*